# Clinical Recommendations for Managing Genitourinary Adverse Effects in Patients Treated with SGLT-2 Inhibitors: A Multidisciplinary Expert Consensus

**DOI:** 10.3390/jcm13216509

**Published:** 2024-10-30

**Authors:** Juan J. Gorgojo-Martínez, José L. Górriz, Ana Cebrián-Cuenca, Almudena Castro Conde, María Velasco Arribas

**Affiliations:** 1Department of Endocrinology and Nutrition, Hospital Universitario Fundación Alcorcón, Alcorcón, 28922 Madrid, Spain; 2Department of Nephrology, Valencia Clinic University Hospital, Instituto de Investigación Sanitaria (INCLIVA), Universitat de València, 46010 Valencia, Spain; jlgorriz@gmail.com; 3Health Centre Casco Antiguo Cartagena, Primary Care Research Group, Biomedical Research Institute of Murcia (IMIB), 30201 Cartagena, Murcia, Spain; anicebrian@gmail.com; 4Department of Cardiology, University Hospital La Paz, IdiPAZ, Biomedical Research Center-Cardiovascular Diseases (CIBERCV-ISCIII), 28046 Madrid, Spain; almudenacastroc@gmail.com; 5Department of Infectious Diseases, Research Department, Hospital Universitario Fundación Alcorcón, 28922 Madrid, Spain; mvelascoa@salud.madrid.org

**Keywords:** SGLT-2 inhibitors, genitourinary infection, genital mycotic infection, urinary tract infection

## Abstract

**Background:** SGLT-2 inhibitors (SGLT-2is) are considered to be a first-line treatment for common conditions like type 2 diabetes, chronic kidney disease, and heart failure due to their proven ability to reduce cardiovascular and renal morbidity and mortality. Despite these benefits, SGLT-2is are associated with certain adverse effects (AEs), particularly genitourinary (GU) events, which can lead to treatment discontinuation in some patients. Preventing these AEs is essential for maintaining the cardiorenal benefits of SGLT-2is. **Methods:** A multidisciplinary panel of experts from various medical specialties reviewed the best available evidence on GU AEs associated with SGLT-2i therapy. The panel focused on the prevention and management of genital mycotic infections, urinary tract infections, and lower urinary tract symptoms in both the general population and high-risk groups, such as renal and cardiac transplant recipients. **Results:** The panel found that permanent discontinuation of SGLT-2is results in a rapid loss of cardiorenal benefits. Preventive strategies, including identifying high-risk patients before initiating therapy, are critical for minimizing GU AEs. Clinical trials show that most GU infections linked to SGLT-2i therapy are mild to moderate in severity and typically respond to standard antimicrobial treatment, without the need for discontinuation. **Conclusions:** Routine discontinuation of SGLT-2is due to GU AEs is not recommended. Therapy should be resumed as soon as possible, unless severe or persistent conditions contraindicate their use, in order to preserve the significant benefits of SGLT-2is in reducing cardiovascular and renal events

## 1. Introduction

Sodium-glucose cotransporter type 2 inhibitors (SGLT-2is) have revolutionized diabetes care, providing not only glycemic control, but also significant cardiovascular (CV) and renal benefits that surpass those of traditional glucose-lowering therapies [1]. These medications lower blood glucose, enhance insulin sensitivity, and promote weight loss by increasing urinary glucose excretion, making them particularly beneficial for individuals with insulin resistance or metabolic syndrome [1]. SGLT-2is also exhibit CV pleiotropic effects, including direct metabolic changes in the myocardium, blood pressure (BP) reduction, and diuretic and natriuretic properties, which alleviate fluid overload and reduce cardiac strain. Additionally, SGLT-2i slower intraglomerular pressure, reduce glomerular hyperfiltration, and decrease albuminuria—a marker of kidney damage—thus helping to preserve kidney function over time, independent of their glucose-lowering effects [1,2,3,4].

Numerous randomized clinical trials (RCTs) have highlighted the significant CV benefits of SGLT-2is in patients with type 2 diabetes mellitus (T2DM), chronic kidney disease (CKD), and heart failure (HF). The EMPA-REG OUTCOME study [2], published in 2015, was groundbreaking, as it was the first to demonstrate a reduction in major adverse CV events (MACEs), such as myocardial infarction, stroke, and CV death, in patients with T2DM and established CV disease (CVD). Similar results were later observed with canagliflozin in the CANVAS program [3] and dapagliflozin in the DECLARE-TIMI 58 study [4] in patients with T2DM, both with and without CVD. Moreover, SGLT-2is have been shown to reduce the risk of hospitalization for HF, regardless of diabetes status, prior HF diagnosis, or ejection fraction, as demonstrated in several RCTs, including the SCORED (sotagliflozin) [5], DAPA-HF (dapagliflozin) [6], DELIVER (dapagliflozin) [7], EMPEROR-Reduced (empagliflozin) [8], and EMPEROR-Preserved (empagliflozin) studies [9]. Additionally, SGLT-2is have shown promise in acute HF settings. The EMPULSE trial [10] demonstrated empagliflozin’s clinical benefits in reducing CV mortality, HF hospitalizations, and improving quality of life in patients hospitalized for acute HF, irrespective of their ejection fraction or T2DM status. Although the EMPACT-MI study [11], which initiated empagliflozin shortly after acute myocardial infarction, did not show a significant reduction in the combined incidence of all-cause death and first HF hospitalization, secondary analyses suggested potential reductions in both first and total HF hospitalizations with empagliflozin.

SGLT-2is have also been demonstrated to slow CKD progression in patients with and without T2DM. Specifically, the CREDENCE study [12] highlighted the significant renal benefits of canagliflozin in patients with advanced CKD and T2DM, while the DAPA-CKD [13] and EMPA-KIDNEY trials [14] showed similar benefits with dapagliflozin and empagliflozin, respectively, in patients with CKD, regardless of their T2DM status.

Overall, these findings underscore the paradigm shift in the treatment of T2DM, CKD, and HF with the advent of SGLT-2is, which offer substantial CV and renal benefits beyond glucose control. However, SGLT-2is are sometimes temporarily or permanently discontinued due to adverse effects (AEs), depriving patients of their metabolic CV and renal benefits.

Few evidence-based consensus documents specifically address the prevention and management of the most common AEs associated with SGLT-2is, and those that do exist are typically not multidisciplinary [15,16]. This review, developed by a panel of experts from various medical specialties with wide experience in SGLT-2i management, aims to evaluate the available evidence and provide practical recommendations for clinicians. The ultimate objective is to reduce the number of patients in real-world settings who miss out on the cardiorenal benefits of these medications due to AEs.

## 2. Materials and Methods

A multidisciplinary panel of experts, including a primary care physician, an internist specializing in infectious diseases, a cardiologist, a nephrologist, and an endocrinologist—each with clinical experience in SGLT-2i therapy across various clinical settings—held virtual meetings to reach a consensus. The panel conducted an extensive literature search using PubMed, prioritizing RCTs, systematic reviews, and meta-analyses. In cases where no RCTs were available, high-quality observational studies or international expert recommendations were used, supplemented by the authors’ real-world clinical experience.

The experts agreed that preventing and managing genitourinary (GU) AEs should be the primary focus when optimizing care for patients receiving SGLT-2is, in order to avoid unnecessary treatment discontinuations that could have significant medium- to long-term clinical consequences. The panel also extended these recommendations to patients without T2DM, given the emerging evidence supporting the use of SGLT-2is in patients with HF or CKD.

Finally, the experts emphasized the need for a dedicated section addressing high-risk patients who are more susceptible to GU infections when using SGLT-2is, ensuring that the best available evidence is applied to this patient population, who may also benefit from this therapeutic class.

## 3. Risks Associated with SGLT-2i Withdrawal

The permanent discontinuation of SGLT-2i therapy may not only lead to worsening glycemic control, but also result in the loss of the cardiometabolic and renal benefits provided by these medications. In the DELIGHT study, after 24 weeks of dapagliflozin treatment in patients with T2DM and CKD, a 3-week follow-up period without medication revealed an almost complete reversal of the drug’s effects on fasting plasma glucose, weight, systolic BP, albuminuria, and eGFR [17]. Similarly, an increase in eGFR one month after the discontinuation of empagliflozin was observed in patients with T2DM and established CVD enrolled in the EMPA-REG OUTCOME study, reflecting the loss of its effects on tubular–glomerular feedback and glomerular hyperfiltration [18].

In a joint sub-analysis of the EMPEROR-Reduced and EMPEROR-Preserved trials in patients with HF, 6799 participants randomized to placebo or empagliflozin 10 mg/day for 1 to 3 years were prospectively evaluated 30 days after treatment withdrawal [19]. Compared to those in the placebo group, patients previously treated with empagliflozin showed a higher risk of CV death or hospitalization for HF, along with greater clinical deterioration, as measured by the KCCQ-CSS score. Additionally, the discontinuation of empagliflozin was associated with significant increases in fasting glucose, body weight, systolic BP, eGFR, NT-proBNP, and uric acid, as well as a significant decrease in hematocrit. These changes were opposite to those observed during the active treatment phase with the drug in the same cohort.

These findings suggest that the abrupt cessation of SGLT-2i therapy, even for short periods, can have serious consequences due to the rapid loss of their beneficial pharmacological effects. Therefore, clinicians should carefully weigh the decision to discontinue SGLT-2is due to AEs, and discontinuation should only occur when the risks posed by AEs outweigh the overall cardiometabolic and renal benefits provided by SGLT-2is, as well as the reductions in morbidity and mortality achieved by continuing treatment.

## 4. Main Causes of SGLT-2i Withdrawal

Withdrawal from SGLT-2i treatment can be temporary or permanent (Table 1) [16,20,21,22,23]. Temporary discontinuation is sometimes necessary prior to scheduled surgery or during prolonged, intense physical activity (e.g., long-distance running) to reduce the risk of diabetic ketoacidosis (DKA). Likewise, temporarily pausing treatment is often recommended when a patient experiences acute illness leading to reduced oral intake or is hospitalized for a medical or surgical condition until the acute phase resolves. Other indications for temporary interruption include an initial eGFR decline of more than 30%, which requires the correction of hypovolemia, or the occurrence of genital mycosis with significant clinical manifestations, in which case, treatment is resumed after the infection is resolved with antifungal therapy. However, the permanent discontinuation of SGLT-2i therapy may deprive patients of the drug’s metabolic and cardiorenal benefits, potentially increasing morbidity and mortality.

The AEs most frequently leading to SGLT-2i discontinuation in RCTs are summarized in Table 2 [2,3,4,12,24,25,26,27,28,29,30,31,32]. In the four major CV safety studies of SGLT-2is, the frequency of AEs leading to treatment discontinuation was similar between the active treatment and placebo groups, as follows: CANVAS (canagliflozin): 35.5 vs. 32.8 participants with an event per 1000 patient years; DECLARE (dapagliflozin): 8.1% vs. 6.9%; EMPA-REG OUTCOME (empagliflozin): 17.3% vs. 19.4%; and VERTIS CV (ertugliflozin): 7.3% vs. 6.8% [2,3,4,32]. Genital mycotic infections (GMIs) are definitely the leading cause of SGLT-2i withdrawal due to AEs, as seen in both RCTs and real-world observational studies [33,34,35]. Nonetheless, the absolute increase in the risk of AEs associated with SGLT-2i use remains lower than the absolute risk reductions in CV and renal morbidity and mortality offered by these drugs [36].

## 5. AES Most Frequently Associated with SGLT-2is

A systematic review and network meta-analysis of 816 RCTs, involving 471,038 patients and evaluating 13 different drug classes for T2DM, confirmed that SGLT-2is significantly increase the risk of GMIs (OR 3.30) and DKA (OR 2.07) compared to other therapeutic classes [37]. These AEs occur less frequently in individuals without diabetes [38,39].

In a recent meta-analysis of nine SGLT-2is across 113 RCTs versus placebo, including 105,293 adult patients, statistically significant increases in GMIs (4.5% vs. 1.0% with placebo) and polyuria (2.7% vs. 0.8% with placebo) were observed [40]. The incidence of other AEs did not differ significantly between SGLT-2is and placebo, as follows: hypovolemia (2.4% vs. 2.0%), renal insufficiency (1.7% vs. 1.2%), acute renal failure (2.0% vs. 2.2%), urinary tract infections (UTIs) (6.8% vs. 5.1%), fractures (3.4% vs. 3.3%), DKA (0.3% vs. 0.1%), amputations (1.6% vs. 1.4%), and severe hypoglycemia (1.9% vs. 1.9%).

When restricted to large CV and renal safety trials, the risk of certain AEs, including GMIs (RR 3.75), volume depletion (RR 1.14), and DKA (RR 2.57), significantly increased compared to placebo [36,41]. However, the association between SGLT-2i use and other AEs, such as amputations, fractures, and UTIs, remains inconsistent across published meta-analyses [36,41,42]. The controversial increase in amputations observed with canagliflozin in the CANVAS trial has not been corroborated by other RCTs or meta-analyses specifically evaluating this AE [43]. The safety outcomes identified in RCT meta-analyses have also been confirmed in real-world cohort studies [44,45]. A cohort study of patients with HF found no significant difference in the risk of UTIs or GMIs among those who initiated treatment with canagliflozin, dapagliflozin, or empagliflozin [46].

Other notable AEs, while not life-threatening, can be bothersome. Lower urinary tract symptoms (LUTSs), including polyuria, nocturia, urinary frequency, and urgency, are common due to the diuretic and natriuretic effects of SGLT-2is [16]. The degree of polyuria tends to be greater in individuals with more pronounced hyperglycemia, although this effect is typically transient. In some patients, particularly those on high-dose diuretics, these effects can lead to volume depletion and hypotension, which may be more severe in patients with diabetes and autonomic neuropathy [47]. These AEs are less common in individuals without diabetes [13].

### 5.1. Genital Mycotic Infections

The risk of developing GMIs can increase up to fivefold in patients treated with SGLT-2is, with the greatest risk occurring within the first month of treatment and persisting throughout its duration [16,48]. GMI is the most common AE listed in the prescribing information for the four EMA-approved SGLT-2is, with an incidence rate of 5% or higher [2,3,4,12,24,25,26,27,28,29,30,31]. In RCTs, most GMIs were mild and did not lead to drug discontinuation. However, their proper identification and management are essential to patient care [23]. Glycosuria resulting from diabetes likely creates a favorable substrate for the growth of organisms, particularly *Candida* species, and this effect is further amplified by the pharmacologic glycosuria induced by SGLT-2is. Additionally, *Candida albicans* has unique mechanisms that promote growth in glucose-rich environments, including a glucose-inducible protein that enhances adhesion and impairs phagocytosis by the host immune system [49]. Other species, such as *Candida glabrata*, may also be implicated in GMIs, posing a challenge due to their poor response to azole antifungal agents. Identifying these species is important for tailoring treatment strategies and assessing the potential need to discontinue SGLT-2is [50].

GMIs associated with SGLT-2is are more common in women than in men and in patients with a history of chronic or recurrent GMIs. Poor genital hygiene can exacerbate these infections by further promoting fungal growth, especially in uncircumcised men with diabetes [23]. However, other variables, such as the duration of diabetes, HbA1c levels, BMI, eGFR, and previous urinary issues, have not been found to be associated with an increased risk of infection in observational studies [51].

### 5.2. Urinary Tract Infections

Individuals diagnosed with T2DM are more susceptible to UTIs and recurrent UTIs compared to those without T2DM. A large retrospective cohort study, which included 179,580 individuals with T2DM, found that UTIs were more prevalent among patients with diabetes (9.4% vs. 5.7%), with a higher incidence in women (14% vs. 5%) [52]. Although the exact mechanisms underlying the increased risk of UTIs in diabetes remain only partially understood, several pathogenetic factors likely contribute to this association. These include the following: (1) higher glucose concentrations in the urine, which alter the urinary microenvironment and promote the growth of various uropathogens; (2) impaired immune responses in patients with diabetes, including defects in polymorphonuclear leukocyte function, adhesion, chemotaxis, and phagocytosis, which may reduce the body’s ability to defend against bacterial proliferation; (3) increased bacterial adherence to uroepithelial cells, which correlates with elevated HbA1c levels; (4) diabetic autonomic neuropathy, which can cause GU dysfunction, leading to bladder dysfunction and increasing the risk of UTIs; and (5) patient-related factors such as age, poor metabolic control, and the duration of diabetes, which further elevate the risk of infection [52,53].

While *Escherichia coli* (*E. coli*) is the most common pathogen in uncomplicated UTIs, patients with diabetes are more susceptible to a broader range of uropathogens, including *Klebsiella* spp., *Staphylococcus aureus*, *Enterobacter* spp., *Proteus* spp., *Pseudomonas* spp., Group B *Streptococci*, and *Enterococcus faecalis*, among others [53]. The glucose-rich environment in the urine of patients with diabetes may contribute to an increased virulence of certain pathogens, such as *Proteus* spp., which utilize glucose as a primary nutrient source. Conversely, studies indicate that *E. coli* primarily relies on peptides and amino acids for growth in the urinary tract, with glucose playing a less central role in its metabolism [54]. Under the nutrient-limited conditions of the urinary system, *E. coli* has adapted to utilize these smaller molecules through various metabolic pathways, such as peptide import and amino acid catabolism, in order to meet its carbon and nitrogen needs. In contrast, the Proteus species, another common uropathogen, is known to utilize glucose more efficiently as a nutrient source under these conditions [55]. This distinction is important, as it reflects how different bacterial species adapt to the nutrient restrictions of the urinary environment and how glucose may contribute to the virulence of certain pathogens like *Proteus* spp., but not necessarily *E. coli*. These findings highlight the complexity of bacterial metabolism and its role in UTIs among patients with diabetes, where the presence of glucose in the urine may favor the growth of specific pathogens, while other bacteria like *E. coli* thrive on different nutrient sources.

Since SGLT-2is increase glucose availability in the urinary tract, they provide a potential substrate for bacterial growth, raising concerns about their association with GU tract infections [56]. While SGLT-2is have been consistently associated with an increased risk of GMIs, the link between these drugs and bacterial UTIs remains unclear, with previous publications reporting conflicting findings. The incidence of UTIs in patients receiving SGLT-2is is significantly lower than that of GMIs, ranging from 3% to 9%. The incidences of both complications, as reported in prescribing information, are presented in Table 2. However, real-world studies have communicated a lower-than-expected prevalence of UTIs [57]. In these studies, UTIs were typically mild to moderate and rarely resulted in treatment discontinuation. In an analysis of 1663 patients, where the reasons for dapagliflozin discontinuation were evaluated, UTIs accounted for only 0.45% of cases [33].

A recent analysis of the FDA Adverse Event Reporting System, which included 1714 UTI cases in patients taking SGLT-2is, suggested an increased risk of UTIs [58]. However, most meta-analyses and RCTs have not demonstrated a significant association between SGLT-2i use and UTI risk [3,4,12,13,18,36,40,41,59,60,61,62,63,64,65] For instance, a meta-analysis of multiple RCTs involving over 50,000 individuals did not show an increased risk of UTIs with SGLT-2i use compared to placebo [60] (Table 3). Similarly, meta-analyses and retrospective cohort studies comparing the UTI risk between SGLT-2is and active comparators did not detect a significant increased risk of UTIs in patients receiving SGLT-2is [3,4,12,13,18,36,40,41,59,60,61,62,63,64,65].

A possible explanation for the lack of real-world evidence for increased clinically significant UTIs, despite glycosuria and the favorable environment for bacterial growth, may be related to the increased urinary flow rate resulting from the osmotic diuresis and natriuresis effects of SGLT-2is. This diuretic effect may reduce bacterial loads and/or prevent the ascension of bacteria within the urinary tract [49]. Nevertheless, anecdotal reports of serious complications such as urosepsis have been noted for patients receiving SGLT-2is, particularly in cases of urinary tract obstruction or an abnormal urinary flow [66]. It is important to consider these risks in patients with bladder outlet obstruction prior to prescribing SGLT-2is.

In summary, current data from multiple real-world studies and meta-analyses suggest no increased risk of clinically significant UTIs with SGLT-2i use. The discontinuation of SGLT-2is due to mild UTI-related complications may deprive patients of the significant CV and renal benefits that these drugs provide. Although a clear association between SGLT-2i use and UTIs has not been established, clinicians should remain cautious when prescribing these drugs to patients with serious or recurrent urogenital infections, an abnormal urinary flow (e.g., incomplete bladder emptying with urinary stasis), or indwelling Foley catheters.

### 5.3. Pollakiuria and Nocturia

SGLT-2is exert their antihyperglycemic and diuretic effects by inhibiting the reabsorption of sodium and glucose in the proximal tubule, leading to natriuresis and osmotic diuresis. These effects can potentially cause LUTSs, which include urinary frequency, urgency, polyuria, and nocturia. This condition, known as overactive bladder, is more common in older adults and individuals with diabetes, especially those with poor metabolic control and/or autonomic neuropathy [67]. Although these new-onset LUTSs are typically mild, they can significantly impact quality of life by increasing urinary frequency and causing sleep disturbances [68].

Any intervention that increases urine output, including SGLT-2is, can exacerbate LUTSs. While no definitive mechanism has been established for these findings, it is speculated that natriuresis and glycosuria-induced osmotic diuresis may contribute to symptoms like pollakiuria and nocturia. The increase in urine output after starting SGLT-2i treatment is transient, typically returning to baseline between the second and fifth day of treatment, with an average increase of about 267 mL/day in people with diabetes [69]. This transient effect may occur due to compensatory sodium reabsorption by other nephrons following the initial natriuresis triggered by SGLT-2is. Additionally, SGLT-2i-induced reductions in plasma volume and BP activate the sympathetic nervous system and the renin–angiotensin–aldosterone system, further increasing sodium reabsorption [70]. Natriuresis following SGLT-2i treatment is dose-dependent, with increases observed 24 h after administration and returning to baseline within 2–15 days, depending on the study [69,70,71].

In individuals without diabetes, SGLT-2 accounts for only about 5% of their total renal sodium chloride reabsorption, but this percentage is much higher in patients with diabetes. This occurs because changes in glucose or sodium filtration modulate SGLT-2 gene expression in the renal proximal tubule’s epithelial cells [72]. This difference may explain why these AEs are less common in individuals without diabetes.

When combined with loop diuretics, SGLT-2is exert synergistic effects. In patients with HF, the coadministration of SGLT-2is and loop diuretics resulted in a fourfold increase in fractional sodium excretion compared to monotherapy [73]. Despite this, SGLT-2is are generally well-tolerated, with few reports of renal function decline, hypokalemia, or hyponatremia [6].

While few studies have examined the effects of SGLT-2is on LUTSs outside of RCTs, one case series involving 50 men with T2DM reported an increase in urinary frequency and nocturia after initiating SGLT-2is, affecting nearly all participants [68]. Therefore, these symptoms may be more significant in real-world settings, where patients are often older than those enrolled in RCTs.

In general, it is not necessary to recommend an increased fluid intake after initiating SGLT-2i therapy, as the diuresis is typically transient. However, it is crucial to inform patients about this potential effect. In some cases, loop diuretic doses may need to be reduced upon the initiation of SGLT-2is to avoid excessive diuresis. Conversely, encouraging additional fluid intake could be harmful, as it may exacerbate LUTSs and lead to drug discontinuation or noncompliance. Identifying the patients most likely to develop LUTSs after starting SGLT-2is—such as older adults, individuals with diabetes and poor metabolic control, and those with preexisting LUTSs—is important for providing appropriate counseling and making necessary adjustments to their treatment. The CV and renal benefits of SGLT-2is definitely outweigh the mild AEs some patients may experience.

## 6. Prevention of GU AEs Associated with SGLT-2is

As previously mentioned, the most common AEs associated with SGLT-2is are GMIs. While UTIs are not definitively linked to SGLT-2is, they may still occur during therapy, and are often cited as a reason for discontinuation. Healthcare professionals should counsel patients receiving these medications on preventive measures, inform them about the clinical manifestations of GU infections, and provide guidance on appropriate actions if they occur. The prescribing information for SGLT-2is recommends monitoring for signs and symptoms of UTIs and GMIs, with prompt treatment if necessary [74,75].

In a real-world study, 125 patients with T2DM who initiated SGLT-2i therapy were given hygiene advice [76]. The incidence of GMIs in these patients was compared with that of 125 individuals who did not receive hygiene advice. The study showed a significant reduction in the risk of GMIs among patients who received hygiene guidance (4.8% vs. 40.8% over 6 months, *p* = 0.015), and compliance was significantly higher in this group.

### 6.1. Urinary Tract Infections

The following practices can help to maintain genital health and reduce the risk of infections and discomfort (Figure 1): [74,75,76,77,78].

Stay hydrated—drinking plenty of fluids helps to maintain urinary tract health by increasing urine output, which dilutes the urine and assists in flushing out bacteria.

Avoid holding urine—delaying urination can lead to bacterial growth in the bladder, increasing the risk of recurrent UTIs. It is important to empty the bladder regularly.

Urinate before and after sexual activity—this simple habit can reduce the likelihood of UTIs by up to 80%.

Evaluate contraceptive methods—some contraceptives, such as spermicides or certain types of condoms, may increase the risk of UTIs. If a woman is prone to UTIs, she should consult her gynecologist about alternative methods that may be better suited for her.

Ensure proper lubrication during sexual activity—insufficient lubrication can lead to vaginal irritation, which may increase the risk of infection. Using a high-quality lubricant can help to prevent this.

Avoid irritating intimate products—overly perfumed or irritating hygiene products can disrupt the natural pH balance, increasing vulnerability to infections. Instead, gentle, pH-balanced products should be opted for, and their use should be limited to once daily.

Favor showers over baths—while an occasional bath is fine, showers are recommended for daily hygiene to reduce the risk of introducing bacteria into the genital area.

Address constipation—chronic constipation can contribute to UTIs by allowing bacteria to colonize areas near the urinary tract. Managing constipation and not delaying bowel movements can help to reduce this risk.

Choose appropriate clothing—cotton underwear should be opted for and overly tight clothing should be avoided, which can create a moist environment conducive to bacterial growth.

Avoid tampons if prone to UTIs—tampon use, particularly during the premenstrual period, has been associated with an increased incidence of UTIs. If this is a concern, alternatives such as pads or menstrual cups, or changing tampons more frequently, should be considered.

### 6.2. Genital Infections

To help prevent these infections in patients using SGLT-2is, healthcare professionals may recommend the following recommendations (Figure 2 and Figure 3): [76,78,79,80,81,82,83].

#### 6.2.1. Daily Intimate Hygiene for Females

Cleanse with water and mild soap—the intimate area should be washed twice daily—once in the morning and once before bed. The vulva should be cleaned thoroughly, including the folds of the labia majora and minora and around the clitoris.

Keep the area dry—maintaining dryness is critical in preventing infections. The area is prone to moisture accumulation due to contact with urine, sweat, vaginal discharge, menstruation, and limited ventilation, making it susceptible to microbial growth.

Wear breathable underwear—cotton underwear should be chosen and tight-fitting or synthetic fabrics should be avoided.

Change immediately after physical activities—after exercising or swimming, people should dry themselves thoroughly and change into clean, dry underwear.

Avoid perfumes and antiseptic—perfumes or antiseptics should not be applied to the intimate area unless prescribed by a healthcare professional.

Change menstrual products regularly—during menstruation, pads or tampons must be changed frequently to maintain hygiene.

#### 6.2.2. Daily Intimate Hygiene for Males

Wash daily with soap and water—the genital area should be cleansed daily, while avoiding using sponges or gloves, which can harbor contaminants like fungi.

Wipe after urination—clean paper should be used to wipe the genital area after urinating.

Thoroughly clean the area—cleansing should be performed from the anus to the penis, groin, testicles, and scrotum, then rinsed thoroughly to avoid irritation.

Uncircumcised men should clean under the foreskin—the foreskin should be gently pulled back and the glans should be washed to ensure proper hygiene.

Avoid creams, deodorants, or powders—these should only be used if prescribed for irritation, injury, or dryness. Products recommended by a healthcare professional should always be used.

Dry thoroughly—the area must be completely dry after washing to prevent moisture buildup, which fosters fungal and bacterial growth. A towel dedicated exclusively for this purpose can be used.

If patients experience any signs of infection—such as irritation, itching of the external genital area, burning during urination, painful intercourse, swelling of the vulva, or unusual discharge with an unpleasant odor—they should consult their healthcare provider immediately.

## 7. Treatment of GU AEs Associated with SGLT-2is

### 7.1. Genital Infections

Patients with certain risk factors are more susceptible to fungal GU infections, and screening for infections should be part of every appointment. Risk factors, other than SGLT-2is, include a history of genital infections, obesity (especially in postmenopausal women), uncircumcised men, and concurrent use of sulfonylurea or insulin. Patients experiencing infection or undergoing antibiotic treatment should not start SGLT-2i therapy until the infection has resolved.

Asymptomatic candiduria should not be treated unless specific conditions are present. Antifungal treatment is recommended for neutropenic patients and those undergoing urinary tract procedures. There is no evidence to support treating asymptomatic candiduria in patients on SGLT-2i therapy. Addressing predisposing factors may help to prevent further infections [23,58,84,85,86].

Most genital infections associated with SGLT-2i therapy are mild to moderate and respond to standard antimicrobial treatment, without the need to discontinue therapy (Figure 4) [58,84,86]. Prompt treatment is associated with better outcomes, and reinforcing perineal hygiene significantly reduces the incidence of genital infections in patients on SGLT-2is [15,23,87]. Patients can generally expect improvement in 24 to 48 h. Complete resolution may take one to two weeks. Routine discontinuation of SGLT2is in the setting of GU infections is not recommended. Patients with fungal pyelonephritis require hospitalization, and surgical treatment is indicated cases involving fungal balls. This severe condition necessitates the discontinuation of SGLT-2is, which should also be avoided in the future [88].

Genital candidiasis should be treated with standard antifungal therapy. Since most safety studies report infection onset within the first 3–6 months of SGLT-2i therapy, maintaining a high clinical suspicion and ensuring rapid management during this period are recommended [89]. Clinical trial data with canagliflozin suggest that the frequency of genital infections decreases over time, with the lowest incidence observed after 21 months of therapy [90,91]. Additionally, there is no clear dose–response relationship between SGLT-2is and genital infections [89].

Genital infections associated with SGLT-2is are classified as sporadic if they occur fewer than three times per year [87,91,92]. When treating these sporadic episodes, several key factors must be considered, as follows: patient preferences, allergies and tolerance, preferred treatment duration, prior response history, drug interactions, the patient’s pregnancy status, the presence of mild to moderate symptoms, the likelihood of *Candida albicans* infection (responsible for approximately 90% of cases), the patient’s immunocompetence, availability, and cost [79,92,93,94,95].

#### 7.1.1. Acute Episodes in Women

Treatment is recommended for women with symptomatic fungal vaginal infections. Asymptomatic women identified through screening, such as Pap smears, do not require treatment. Culture sampling is generally unnecessary for a first episode unless there is suspicion of other infections, such as trichomoniasis or bacterial vaginosis [88,92]. Most vulvovaginal candidiasis (VVC) cases are caused by *Candida albicans*, with *Candida glabrata* and *Candida krusei* being less common.

Treatment should be individualized, as previously noted. Uncomplicated VVC can be managed with oral azoles, topical azoles, or triterpenoid drugs (Table 4, Figure 4), with no single regimen demonstrating superiority [84,87,88,92,94]. A single 150 mg dose of oral fluconazole may be considered as the first-line option for patients with poor adherence to therapy, a preference for oral treatment, a history of poor response to topical therapies, or moderate-to-severe infection. In cases of severe vaginitis, oral fluconazole at 150 mg every 72 h for two or three doses, depending on severity, is recommended. A low-potency topical corticosteroid can be applied to the vulva for up to 48 h to alleviate symptoms until the antifungal medication takes effect [88].

#### 7.1.2. Acute Episodes in Men

*Candida* balanitis, the most common identifiable cause of balanitis, occurs more frequently in men with diabetes, obesity, a history of sexually transmitted infections, and those who are uncircumcised [95,96].

No studies on patients receiving SGLT-2is have demonstrated a superior efficacy of any specific topical or oral regimen. Therefore, treatment choice should be reached in consultation with the patient. First-line treatment includes topical antifungal agents or a single 150 mg dose of oral fluconazole, following the same criteria used for women (Table 4, Figure 4). Most RCTs involving SGLT-2is have shown that discontinuing therapy during an episode of genital infection does not improve prognosis [81].

Clotrimazole 1% (applied once daily for 7 days) or 2% (applied once daily for 5 days) or miconazole 2% (applied twice daily) can be used. Nystatin cream (100,000 units/g for 7 days) is an alternative for patients allergic to imidazoles. For severe symptoms, oral fluconazole every 72 h for two or three doses or a combination of topical imidazole and hydrocortisone 1% cream applied twice daily may be considered. In uncircumcised patients with nonspecific balanitis, saline solution baths may be effective. Alternative therapies, such as dietary supplementation with lactobacillus-containing yogurt, are not recommended [92,93,94]. Female sexual partners of patients with balanitis should be offered testing for Candida or empiric treatment to reduce the likelihood of reinfection.

### 7.2. Acute UTIs

#### 7.2.1. General Recommendations

Patients receiving SGLT-2is, particularly those with diabetes, should be informed about the potential risk of UTIs and advised to seek medical attention if symptoms arise [58,84,94,97].

Routine urine cultures and prophylactic or preemptive antimicrobial therapy are not recommended for preventing UTIs in asymptomatic patients initiating SGLT-2is [79,97]. Treating asymptomatic bacteriuria does not reduce the risk of complicated infections and may contribute to increased antibiotic resistance [85,94,98].

Conversely, urine cultures should be obtained in symptomatic patients—those presenting with dysuria, urinary frequency, urgency, suprapubic pain, abdominal or lumbar pain, or fever—who are undergoing SGLT-2i treatment [85,94,99]. Empiric antimicrobial therapy should be initiated in symptomatic patients and adjusted based on urine culture and susceptibility test results, which are generally available within 48 h [94,99,100]. The choice of empiric therapy should be based on the severity of illness, risk factors for resistant pathogens, and patient factors such as allergies and prior antimicrobial use (Table 5 and Figure 5) [99].

Patients in RCTs taking SGLT-2is who developed a UTI did not stop taking the blinded assigned study drug and there was no increased risk of UTI severity or recurrent UTI compared to placebo [15]. Temporary discontinuation of the drug should, therefore, be individualized and discussed with the patient.

#### 7.2.2. Treatment of Outpatients

Patients with acute complicated UTIs of mild to moderate severity who can reliably take oral medications may be treated on an outpatient basis.

Empiric antibiotic treatment—for patients without multidrug-resistant (MDR) infection risk, the antibiotic should be selected based on local resistance patterns. Beta-lactam antibiotics, trimethoprim–sulfamethoxazole, and fluoroquinolone-based regimens are effective, but should be avoided in patients with allergies or intolerance. Fluoroquinolones should not be used in patients with a prolonged QT interval or other risk factors for torsades de pointes, and trimethoprim–sulfamethoxazole should be avoided in cases of renal failure. Fosfomycin-tromethamine is an effective oral option for a first episode. Treatment options are listed in Table 5 [94,99].

Guided antibiotic treatment—urine culture and susceptibility results should be monitored to confirm the appropriateness of the empiric antimicrobial regimen and guide any necessary adjustments. The treatment duration is typically 3–7 days, depending on the clinical syndrome and the antibiotic chosen (see Table 5). Extended durations or further evaluation may be required for patients with delayed improvement [94,99].

#### 7.2.3. Hospitalization Indications

Patients with severe symptoms, high fever, low BP, marked debility, or suspected urinary tract obstruction should be hospitalized. Outpatients with mild to moderate symptoms can be treated with oral medications, considering the risk factors for MDR infections [94,99]. These risk factors include prior MDR colonization, recent antibiotic use (within the past month), anatomical abnormalities, urological manipulation, or healthcare-associated infections [99].

### 7.3. The Challenge of Resistance to Antibiotic and Antifungal Therapy

Antibiotic and antifungal resistance poses a significant challenge in clinical management. Preventing the emergence of resistance through the proper treatment of acute infections, reducing recurrences, and addressing modifiable risk factors is essential. The use of antifungal and antibiotic therapies guided by susceptibility testing, along with consultation from infectious disease specialists for multidrug-resistant species, can improve outcomes in managing resistant infections [100].

Resistance patterns should be monitored regularly, with adjustments made to treatment regimens based on susceptibility testing results. In some cases, combining antifungal agents or rotating therapies may help to manage resistant infections. Reducing unnecessary antibiotic use and adhering to evidence-based guidelines are critical strategies for mitigating resistance development [101,102,103,104].

Non-albicans Candida species, particularly *C. glabrata* and *C. krusei*, present significant therapeutic challenges due to their inherent resistance to common antifungal agents and the diversity of species involved. Guided antifungal therapy is recommended to enhance effectiveness, minimize treatment burden, and reduce toxicity risks [50,105,106].

#### 7.3.1. Assessment and Diagnosis

-Suspicion—*C. glabrata* or *C. krusei* infection should be suspected in cases of recurrence, slow response to treatment, or poor progression of the episode.-Identification of the causal agent—diagnosis should be confirmed through cultures and susceptibility testing to identify the specific Candida species and its resistance profile.-Exclusion of other causes—before initiating treatment for VVC, it is important to rule out other causes of vaginal symptoms, such as bacterial infections or sexually transmitted diseases.

#### 7.3.2. Treatment for Resistant Candida Species (Table 4)

-Vaginal Boric Acid—administration of 600 mg boric acid capsules vaginally each night for 2 to 3 weeks. This treatment is effective in approximately 70% of patients with confirmed *C. glabrata* infections. However, its use is contraindicated in pregnant women due to limited safety data and the risk of toxicity if ingested [50].-Topical azole—for individuals with *C. krusei* infection, treatment with a topical azole (cream or suppository) is recommended.-Amphotericin B Suppository—a 50 mg vaginal suppository of Amphotericin B, administered for 14 nights, is effective in persistent cases of *C. glabrata* infection [105].-Itraconazole—if topical treatments are ineffective, oral itraconazole can be considered (200 mg twice daily). However, due to its potential toxicity, topical therapies should remain as first-line treatments.-Voriconazole—due to limited efficacy data and its potential hepatic toxicity, voriconazole should only be considered when all other topical and systemic therapies have failed.-Ibrexafungerp—this oral triterpenoid antifungal may be an option for resistant Candida species, though data on its use are limited. Its mechanism of action helps to avoid cross-resistance. Ibrexafungerp is FDA-approved for treating VVC in postmenarchal women and girls, but is not yet approved by the EMA and may not be available in certain countries [107].

#### 7.3.3. Prevention

Preventive treatments, such as oral nystatin or probiotics, have shown limited success [23,92]. Treating sexual partners is not recommended unless recurrent VVC is confirmed. It is also important to avoid potential triggers for recurrence, such as unnecessary antibiotic use, and to discuss the continuation of SGLT-2is with patients in severe cases.

## 8. Treatment of GU AEs Associated with SGLT-2is in High-Risk Patients

### 8.1. Recurrent Infections

#### 8.1.1. Genital Recurrent Infections

Recurrent genital infections are defined as more than three episodes per year. These infections should be thoroughly evaluated using differential diagnosis to distinguish them from conditions such as bacterial vaginosis, trichomoniasis, sexually transmitted infections, or other forms of balanitis [88,92,101,108]. A genital exudate culture is essential to confirm the diagnosis and assess for concurrent infections that may involve multiple organisms requiring different treatments. While initial infections may not require microbiological evaluation, recurrent episodes must be evaluated both clinically and microbiologically. A culture-proven diagnosis with in vitro susceptibility testing is mandatory for recurrent cases [88,101,108].

Recurrent infections can present with severe symptoms, including extensive erythema, edema, and fissures. In males, chronic balanitis may result in ulcerative lesions of the glans and foreskin, and potentially lead to meatal or urethral stricture. Chronic balanitis can also predispose patients to premalignant and malignant lesions. In males with chronic balanitis, careful inspection for phimosis and paraphimosis is essential. Circumcision may be considered when balanitis recurs despite proper hygiene measures and directed treatment [94,95].

Recurrent infections may be associated with persistent risk factors, drug resistance, or inadequate prior treatments. Chronic management involves patient-specific treatment plans considering drug interactions, prior response history, preferred treatment duration, potential side effects, and cost. To date, high-quality evidence for treating recurrent VVC is limited [101]. Given the lack of robust evidence for managing recurrent genital infections in patients taking SGLT-2is, it is essential to discuss the risk–benefit balance of continuing SGLT-2i therapy with the patient. Avoiding SGLT-2is may be considered for female patients with a history of severe, recurrent fungal infections. Patients with balanoposthitis who are already on SGLT-2is need not discontinue treatment. However, if the condition becomes refractory or recurs frequently, switching from SGLT-2is might be considered.

The optimal treatment for recurrent VVC has yet to be defined, and must be individualized [101,108]. Continuous or intermittent antifungal therapy may be necessary. Some treatment regimens are provided in Table 4, assuming that sensitivity is confirmed. Maintenance therapy with fluconazole at 150 mg weekly or itraconazole at 200 mg weekly may help to control recurrences [101]. An extended fluconazole regimen comprises oral induction with 150 mg every 72 h for three doses, followed by an oral maintenance dose of 150 mg once per week for six months. Oral fluconazole can cause gastrointestinal intolerance, headaches, rash, and transient liver function abnormalities, and should not be used in pregnant women. Potential interactions between azole treatments and some drugs, for example, statins, should be revised, although such interactions are not very common at the dose used to treat GMIs.

#### 8.1.2. Recurrent Bacterial UTIs

Recurrent UTIs are defined as more than two episodes in six months or three episodes in twelve months [94,99]. Diagnostic urine testing with urine culture is recommended for each acute episode to guide antibiotic therapy, especially when considering antibiotic prophylaxis, and to differentiate between recurrent and relapsing UTIs (the recurrence of the same uropathogen strain within two to four weeks after treatment) [85,94,99]. Distinguishing between recurrent and relapsing UTIs is crucial, as inadequate treatment of anatomical abnormalities may contribute to relapse [99].

Imaging and urologic evaluations are recommended for men and women with suspected structural or functional abnormalities of the GU tract [99]. Indicators for evaluation include the following: (1) relapsing infection; (2) repeated isolation of Proteus species; (3) history of nephrolithiasis; (4) persistent hematuria; and (5) voiding abnormalities. Patients with diabetes mellitus and recurrent cystitis should also undergo imaging and urologic evaluation [94,99].

A comprehensive risk factor assessment should be conducted. In addition to the classic risk factors for recurrent cystitis—such as urinary anatomical abnormalities, atrophic vaginitis, spermicide use, genetic predisposition, diabetes, and hygiene factors—the role of SGLT-2is should be evaluated. Furthermore, the risks and benefits of continuing these medications must be discussed with the patient, particularly considering their discontinuation in severe cases [84,85,97].

Initial preventive strategies—non-antimicrobial preventive strategies include the following:-Increase fluid intake—if possible, fluid intake should be increased to 2 to 3 L daily to reduce the risk of recurrence [109].-Behavioral changes—while not extensively studied, behavioral changes such as avoiding spermicides and postcoital voiding are reasonable to attempt.-Vaginal estrogen—for postmenopausal women with recurrent cystitis, vaginal estrogen is suggested [110].

Antibiotic prophylaxis—antibiotic prophylaxis is one of the most effective measures [104,111]. Postcoital prophylaxis is used for women with cystitis associated with sexual activity, while continuous prophylaxis is used for other cases. Long-term low-dose antibiotic prophylaxis (at least six months) is the gold standard for preventing recurrent UTIs (Figure 5). The choice of antibiotic should be based on previous uropathogen susceptibility patterns, drug allergies, and potential drug interactions [104,111]. Specific doses and options are outlined in Table 5.

Additional Preventive Interventions—antibiotic-sparing strategies, such as cranberry products (e.g., an 8 ounce glass of cranberry juice once or twice daily or cranberry concentrate tablets at 500–1000 mg daily), may be used, though their efficacy is lower than that of antibiotic prophylaxis [112]. In a small RCT, the daily administration of 120 mg of highly standardized cranberry in postmenopausal women with T2DM taking SGLT-2 inhibitors led to a statistically significant reduction in UTI episodes in the supplemented group in comparison to those with placebo administration [113]. Methenamine hippurate (1 g orally twice daily) was shown to be not inferior to daily low-dose antibiotics in a large pragmatic clinical trial [114,115]. D-mannose did not demonstrate efficacy in a recent clinical trial [116]. There is conflicting evidence regarding the efficacy of probiotics [117,118]. Some vaccines for preventing recurrent UTIs, such as MV140 [119], have shown promising clinical efficacy in small trials.

Reevaluation—preventive strategies should be reevaluated after three to six months to ensure effectiveness and patient adherence. Discontinuing SGLT-2is can be considered if episodes cannot be controlled or are highly symptomatic, after discussing the risk–benefit balance with the patient [58,85,97].

### 8.2. Catheter Use

Patients using catheters are at an increased risk for GU infections. However, a pre–post study involving catheterized patients treated with empagliflozin did not demonstrate an increased risk of UTIs [120]. Management strategies should include rigorous hygiene protocols, tailored antifungal or antibiotic treatments based on the specific infection, and proper catheter care [121].

Asymptomatic bacteriuria is common among catheterized patients. Treating asymptomatic bacteriuria does not improve outcomes and may promote the emergence of resistant bacteria [97,98]. Therefore, screening and treatment for asymptomatic bacteriuria in catheterized patients are generally not recommended. Similarly, asymptomatic candiduria rarely requires antifungal therapy [121]

The selection of empiric antimicrobial therapy for symptomatic episodes should consider the likelihood of resistant infections and the extent of the infection, such as cystitis or more severe conditions like pyelonephritis or prostatitis. Fever is often present in infections beyond the bladder [94,121]. Once culture and susceptibility results become available, the antimicrobial regimen should be tailored to target the specific organism identified [121].

Recent catheter removal may increase the risk of catheter-associated infection. In the absence of fever, patients may be managed as having acute simple cystitis. There is no evidence supporting the use of prophylactic antibiotics before catheter changes [99]. Whenever feasible, bladder catheters or urologic stents should be removed. If complete removal is not possible, replacing the devices or intermittent bladder catheterization may help to reduce colonization [94,121].

### 8.3. Kidney Transplant

Diabetes is a common complication following kidney transplantation. Post-transplant diabetes mellitus (PTDM) is associated with an increased mortality and morbidity, primarily due to higher rates of CVD and infections, which are the leading causes of death in kidney transplant recipients (KTRs) [122]. Over the past decade, significant advances have been made in understanding PTDM, alongside the rapid evolution of treatment algorithms for managing diabetes in the general population.

Since the mechanisms driving renal disease progression are likely similar in patients with T2DM with CKD and in KTRs with diabetes, it seems plausible that SGLT2is could offer unique benefits to kidney transplant recipients with diabetes, proteinuria, or HF, potentially improving both allograft longevity and CV risk. However, these renal benefits are not fully understood in the context of a solitary, denervated kidney [123].

While there is substantial evidence supporting the use of SGLT2is in patients with diabetic kidney disease, particularly for nephroprotection and a reduction in CV and mortality risk, clinical trials focused specifically on kidney transplant patients are lacking [124]. A systematic review of eight studies concluded that, among KTRs with T2DM and excellent kidney function, SGLT2is effectively lowered HbA1c, reduced body weight, and preserved kidney function, with no reported serious AEs such as euglycemic ketoacidosis or acute rejection [125]. In recent years, several new studies on SGLT2i use in KTRs have been published; however, only one was an RCT, and it involved a small number of patients [126,127,128,129,130].

SGLT2is remain underutilized in the management of PTDM, largely due to limited transplant-specific evidence and concerns about increased risks of GMIs and UTIs, both of which are already more common in KTRs. UTIs are the most frequent infectious complication in KTRs, affecting up to 25% of recipients in the first year and accounting for up to 30% of hospitalizations due to sepsis [131]. Their use in KTRs also raises concerns related to the presence of a solitary functioning kidney with abnormal genitourinary anatomy post-surgery, the placement of urinary catheters, concurrent use of maintenance immunosuppression, a high prevalence of immunomodulatory viral infections, and the overall compromised immune status of these patients. Consequently, the recent KDIGO guidelines on CKD and diabetes recommend a cautious approach to adopting SGLT2is in this population [124].

Although no specific studies have assessed the safety of SGLT2is in KTRs, some observational data are available. A Spanish multicenter observational study involving 338 patients from 16 centers evaluated the incidence of UTIs and GMIs in KTRs with T2DM or PTDM following SGLT2i treatment [130]. Over six months, 26% of patients experienced an adverse event, with UTIs being the most common, affecting 14% of individuals. In 10% of cases, SGLT2i treatment was discontinued, primarily due to UTIs. However, a post hoc subgroup analysis found that the incidence of UTIs was similar between KTRs with diabetes treated with SGLT2is over 12 months and KTRs without diabetes (17.9% versus 16.7%).

In the absence of specific guidelines for SGLT2i use in transplant patients, an international PTDM consensus was recently published, which includes recommendations for SGLT2i use in KTRs [122]. Since KTRs are exposed to numerous risk factors for hemodynamic ischemic injury during the immediate and early post-transplant period, including infections, the consensus recommends that SGLT2is can be used to treat PTDM once stable graft function is achieved. Initiation should be guided by comorbidities such as heart failure (favoring use) and significant urosepsis or a high risk of severe GMI (discouraging use), although current studies have not shown an increased risk of UTIs with SGLT2is [122].

While there are no specific recommendations regarding the timing of SGLT2i initiation in KTRs, most patients in published studies began treatment at least one year post-transplant. This suggests a preference for later initiation, likely due to nephrologists’ comfort levels and the potentially lower infection risk in patients further from transplant, as they tend to be on lower doses of immunosuppressants [123]. Earlier use could be considered, but extra caution is warranted, taking into account patient-specific factors such as recent UTI history, surgical requirements, kidney function, and current immunosuppression levels [132]. Large RCTs with longer follow-up periods are needed to assess the long-term cardiovascular and renal outcomes, as well as the safety of SGLT2i therapy in the transplant setting.

### 8.4. Heart Transplant

SGLT-2is may be considered for heart transplant (HT) patients in several scenarios, including pre-existing T2DM, CKD, and the onset of post-transplant diabetes. In these cases, the safety profile and usage recommendations for SGLT-2is remain consistent.

Emerging data on the use of SGLT-2is in HT patients are promising. Both T2DM and post-transplant diabetes are associated with an increased morbidity and mortality. The efficacy and safety of SGLT-2is in transplant recipients are still being investigated. In a retrospective analysis [133], 22 HT recipients treated with empagliflozin were compared to 79 patients receiving other glucose-lowering therapies. In this study, empagliflozin appears to be a safe and effective option for managing select HT patients with diabetes. Minimal AEs were reported with empagliflozin, with only one case requiring treatment discontinuation and no observed GU infections. Over 12 months, empagliflozin resulted in reductions in weight, BMI, HbA1c, and furosemide dose compared to the control group. No significant differences in BP or renal function were observed between the empagliflozin and control groups. In another retrospective analysis [134], the safety and effectiveness of GLP-1RAs and SGLT-2is were evaluated in patients who had undergone orthotopic HT at a high-volume center. Among the 21 patients studied, significant reductions in weight, insulin usage, HbA1c, and LDL-cholesterol were observed. Both SGLT-2is and GLP-1RAs were well tolerated, with no AEs leading to treatment discontinuation. Although larger studies are warranted, this exploratory study suggests that both SGLT-2i and GLP-1RAs are safe and effective therapeutic options for managing T2DM post-HT.

In HT recipients, who are often on immunosuppressive therapy, the risk of GU AEs associated with SGLT-2is, particularly GMIs, may be heightened. A recent review of 20 studies in solid organ transplant recipients receiving SGLT-2is, including four studies in patients with HT, showed that the use of these drugs in this scenario seems to have a similar A1c-lowering effect and rate of AEs compared with the general patient population [132]. The authors conclude that, according to the available data, an SGLT2i could be prescribed in solid organ transplant recipients for diabetes at 1 year after transplant. Patients are on the highest amount of immunosuppression in the months right after transplant; after this initial period, they tend to be on less immunosuppression, hence, they also have a lower risk of infection. Earlier use of SGLT-2is could be considered in special cases, but extra caution surrounding patient-specific factors, including recent UTI history and immunosuppression, would need to be assessed prior to initiation.

The management of GU AEs in these individuals typically follows the same protocols as for other patients, including topical and antifungal medications for GMIs and antibiotics for UTIs, based on the bacteria identified and its antibiotic sensitivity profile and hygiene practices. Close monitoring for GU AEs is essential in HT recipients on SGLT-2is. Regular follow-up visits can facilitate the early detection and prompt management of complications. It is crucial for HT recipients to report any GU symptoms or concerns to their healthcare provider. In some cases, adjusting medication dosages or treatment regimens may be necessary to minimize AEs while still allowing patients to benefit from the CV benefits of SGLT-2is. Although larger safety trials are needed, preliminary findings suggest that SGLT-2is are appropriate for post-HT patients. Given the beneficial non-glycemic effects of SGLT-2is, these agents may become a key component in managing post-transplant diabetes mellitus and other conditions in the future [135].

## 9. Strengths and Limitations

The main strengths of this review are, first, its multidisciplinary and highly practical approach from the perspective of various specialties involved in managing patients who benefit from treatment with SGLT-2is. Additionally, an exhaustive literature search was conducted, prioritizing publications with the highest level of evidence. The primary limitation of this publication is that it was not carried out following a formal clinical practice guideline methodology, so it should be considered merely a consensus, which does not hold the same weight as a guideline. Moreover, the number of high-quality studies evaluating the management of GU infections in patients treated with SGLT-2is is currently very limited.

## 10. Conclusions

SGLT-2is are now considered to be a first-line treatment for prevalent conditions such as T2DM, CKD, and HF due to their well-established benefits in reducing CV and renal morbidity and mortality, as demonstrated in numerous RCTs. Most GU infections associated with SGLT-2i therapy have been shown in clinical trials to be mild to moderate in severity and typically respond to standard antimicrobial treatment, without the need to discontinue therapy. However, some patients permanently stop SGLT-2i treatment due to GU AEs. GMIs are the leading cause of SGLT-2i withdrawal in both RCTs and real-world observational studies. While most meta-analyses and RCTs have not demonstrated a significant association between SGLT-2i use and increased UTI risk, UTIs remain a common reason for SGLT-2i discontinuation.

Notably, there is a lack of high-quality evidence to guide the prevention and management of GU AEs in patients on SGLT-2i therapy. Consequently, our recommendations (Figure 1, Figure 2, Figure 3, Figure 4 and Figure 5) are primarily based on previous clinical trials involving patients with GMIs or UTIs unrelated to SGLT-2i therapy. The most important interventions for minimizing treatment discontinuation involve identifying patients at a high risk for GU infections and implementing preventive strategies prior to initiating therapy.

Based on the favorable outcomes observed in RCTs for patients experiencing UTIs or GMIs who continued treatment, routine discontinuation of SGLT-2is is not recommended when a GU AE occurs, except in severe cases. SGLT-2i therapy should be resumed as soon as possible, unless a severe or persistent condition contraindicates its use, given the risk-benefit profile of this drug class. Discontinuing SGLT-2i therapy may increase cardiorenal risk, as the benefits of these agents can be lost within just a few weeks.

Further epidemiological, mechanistic, and interventional studies on GU infections in patients treated with SGLT-2is are urgently needed to strengthen the current evidence and improve the management of these AEs.

## Figures and Tables

**Figure 1 jcm-13-06509-f001:**
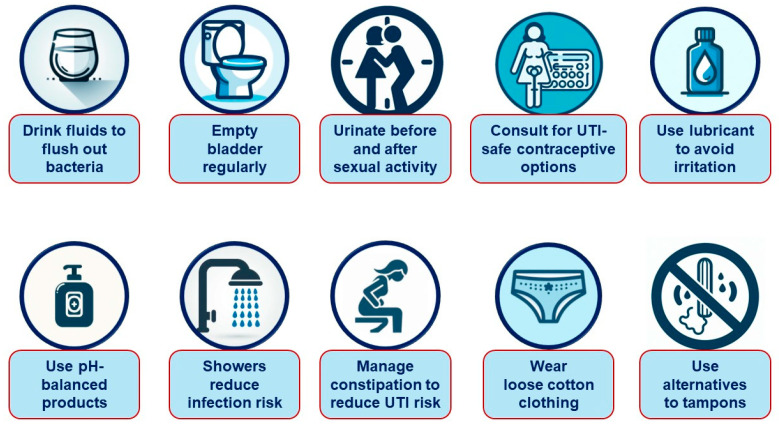
Clinical recommendations for the prevention of urinary tract infections in patients treated with SGLT-2 inhibitors.

**Figure 2 jcm-13-06509-f002:**
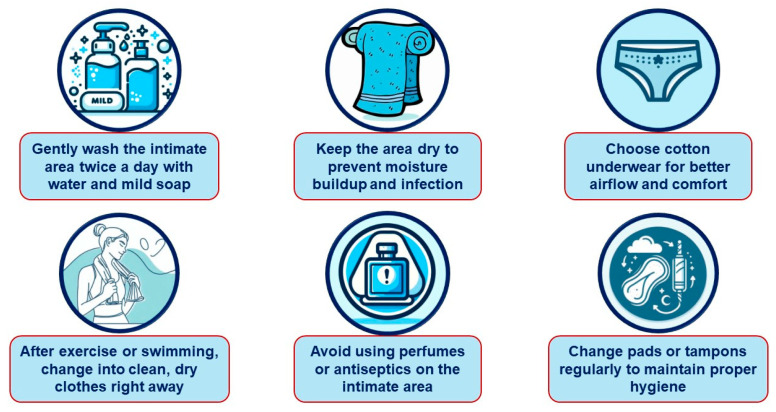
Clinical recommendations for the prevention of genital mycotic infections in women treated with SGLT-2 inhibitors.

**Figure 3 jcm-13-06509-f003:**
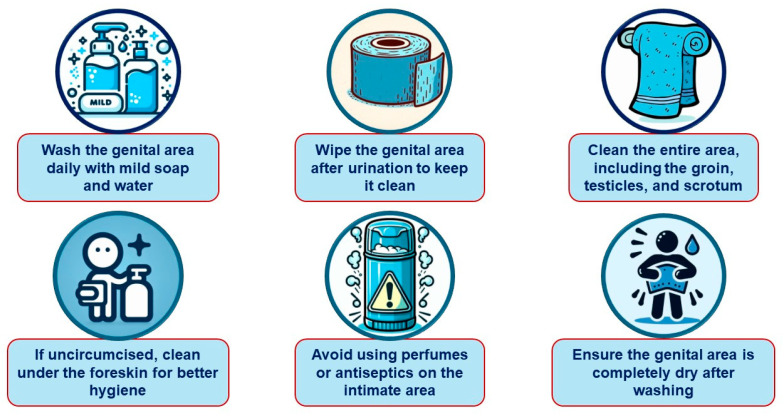
Clinical recommendations for the prevention of genital mycotic infections in men treated with SGLT-2 inhibitors.

**Figure 4 jcm-13-06509-f004:**
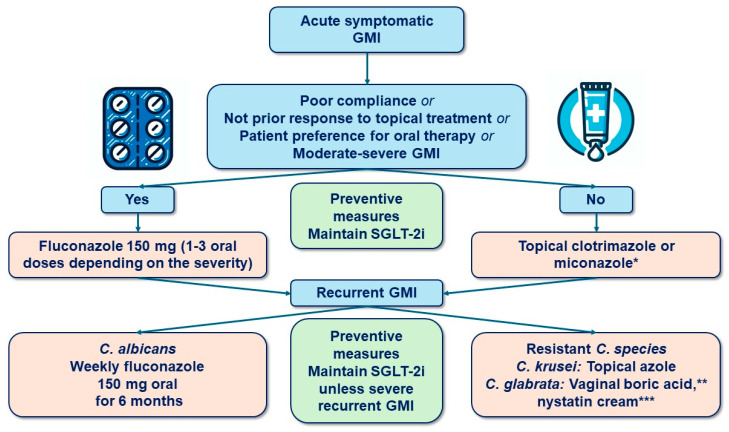
Clinical recommendations for the management of genital mycotic infections in patients treated with SGLT-2 inhibitors. * Plus hydrocortisone 1% cream in balanitis. ** In vulvovaginal candidiasis. *** In balanitis.

**Figure 5 jcm-13-06509-f005:**
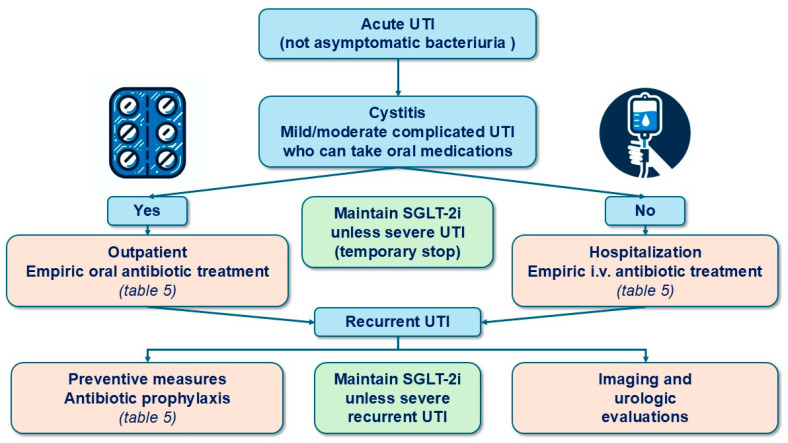
Clinical recommendations for the management of urinary tract infections in patients treated with SGLT-2 inhibitors.

**Table 1 jcm-13-06509-t001:** Causes of SGLT-2i treatment interruption in clinical trials and in real-world studies [16,20,21,22,23].

Temporary Interruption
Scheduled surgery (72 hours before) Intense and prolonged physical activity Acute illness with reduced oral intake Hospitalization for severe medical illness or major surgery Volume depletion Initial drop in eGFR > 30% Severe genital mycosis Pyelonephritis or urinary sepsis Fournier’s gangrene Pregnancy and lactation
2. Permanent Interruption
Recurrent GU infections DKA Increased urinary frequency not tolerated by the patient Initiation of dialysis or renal transplant Diagnostic reclassification from type 2 diabetes to type 1 Hypersensitivity reaction to the drug or the excipients

**Table 2 jcm-13-06509-t002:** Frequency of AEs in RCTs with SGLT-2is. Comparison of maximum dose of SGLT-2is vs. placebo. References [3,4,12,24,25,26,27,28,29,30,31,32]. GMIs: genital mycotic infections. UTIs: urinary tract infections. DKA: diabetic ketoacidosis. NR: not reported.

	Canagliflozin	Dapagliflozin	Empagliflozin	Ertugliflozin
**GMIs**				
Females	11.6% vs. 2.8%	6.9% vs. 1.5%	6.4% vs. 1.5%	12.2% vs. 3.3%
Males	3.8% vs. 0.7%	2.7% vs. 0.3%	1.6% vs. 0.4%	4.2% vs. 0.4%
**UTIs**	4.3% vs. 4.0%	4.7% vs. 3.5%	7.0% vs. 7.2%	4.1% vs. 3.9%
**Volume depletion**	1.3% vs. 1.1%	1.1% vs. 0.7%	0.4% vs. 0.3%	<2% vs. <2%
**Increased urination**	4.6% vs. 0.7%	3.8% vs. 1.7%	3.3% vs. 1.4%	2.4% vs. 1.0%
**DKA**	- CANVAS: 0.6/1000 pat-y vs. 0.3/1000 pat-y	DECLARE: 0.31% vs. 0.14%	EMPA-REG: <0.1% vs. <0.1%	VERTIS-CV: 0.4% vs. 0.1%
**Amputations**	- CANVAS: 2.4% vs. 1.1% - CREDENCE: 3.2% vs. 3.1%	DECLARE: 1.4% vs. 1.3%	EMPA-REG: 1.9% vs. 1.8%	VERTIS-CV: 2.1% vs. 1.6%
**Bone fractures**	- CANVAS: 1.8/100 pat-y vs. 1.1/100 pat-y - CANVAS-R: 1.1/100 pat-y vs. 1.3/100 pat-y - CREDENCE 1.2/100 pat-y vs. 1.2/100 pat-y	DECLARE: 5.3% vs. 5.1%	EMPA-REG: 3.7% vs. 3.9%	VERTIS-CV: 3.7% vs. 3.6%
**Fournier’s Gangrene**	NR	DECLARE: 0.01% vs. 0.06%	NR	VERTIS-CV: 0% vs. 0%

**Table 3 jcm-13-06509-t003:** Studies comparing UTI risk between SGLT-2is and placebo or active comparators. Adapted from [59].

Studies Comparing UTI Risk Between SGLT2is and Placebo			
Comparison	Study [ref.]	Patients (*n*)	Outcome
**Meta-analysis**			
SGLT2is vs. placebo	Puckrin et al [60]	72 trials: 37,116	Risk ratio 1.03; (95% CI: 0.96 to 1.11)
SGLT2is vs. placebo	Johansen et al [36]	8 trials; 49,587	Risk ratio: 1.08; (95% CI: 1.00 to 1.18). *p* = 0.77
SGLT2is vs. placebo	Qiu et al [41]	8 trials 63,237	Risk ratio: 1.09; (95% CI: 0.99 to 1.15). *p* = 0.77
SGLT2is vs. placebo	Liu et al [61]	17 trials: 4,997,145	Risk ratio: 1.29; (95% CI: 1.06 to 1.57) *p* = 0.65
**Randomized controlled trials**			
CREDENCE: Canagliflozin (100 mg) vs. placebo	Perkovic et al [12]	4397	HR 1.08; (95% CI: 0.90 to 1.29)
CANVAS: Canagliflozin (all doses) vs. placebo	Neal et al [3]	4330	40 vs. 37 participants with an event per 1000 patient years; *p* = 0.38
DAPA-CKD: Dapagliflozin (10 mg) vs. placebo	Heerspink et al [13]	4298	No difference reported; details unpublished
DECLARE: Dapagliflozin (10 mg) vs. placebo	Wiviott et al [4]	17,143	HR 0.93; (95% CI: 0.73 to 1.18); *p* = 0.54
EMPA REG OUTCOME: Empagliflozin (all doses) vs. placebo	Wanner et al [18]	7018	eGFR < 60 mL/min per 1.73 m^2^: Rate ratio: 1.06; (95% CI: 0.86 to 1.3) eGFR ≥ 60 mL/min per 1.73 m^2^: Rate ratio 0.92; (95% CI: 0.8 to 1.07)
SGLT2is vs. placebo	Bai et al [62]	3 trials 17,802	HR 1.00; (95% CI: 0.90 to 1.11), *p* = 0.958
**Studies comparing UTI risk between SGLT2is and active comparators**			
**Comparison**	**Study [ref.]**	**Patients (n)**	**Outcome**
**Meta-analysis**			
SGLT2is vs. Active comparator	Puckrin et al [60]	22 trials: 15,966 patients	Random-effects model risk ratio 1.08;(95% CI: 0.93 to 1.25)
**Retrospective cohort**			
SGLT2is vs. GLP1-RA	Varshney et al [64]	474 patients	Composite genitourinary infection HR 0.78; (95% CI: 0.26 to 2.37)
SGLT2is vs. DPP4i	Fisher et al [65]	416,488 patients	Urosepsis HR 0.58; (95% CI: 0.42 to 0.80)
SGLT2is vs. DPP4i or GLP1-RA	Dave et al [63]	SGLT2is vs. DPP4i: n 123,752; SGLT2is vs. GLP1-RA: n 111,978	*Severe UTI:* - SGLT2is vs. DPP4i: HR 0.98; (95% CI: 0.68 to 1.41) - SGLT2is vs. GLP1-RA: HR 0.72; (95% CI: 0.53 to 0.99) *Treated outpatient UTI:* - SGLT2is vs. DPP4i: HR 0.96; (95% CI: 0.89 to 1.04) - SGLT2is vs. GLP1-RA: HR 0.91;(95% CI: 0.84 to 0.99)
SGLT2is vs. active comparator	Li CX et al [40]	40 trials, 9,911,454 patients	HR 0.99; (95% CI: 0.89 to 1.10), *p* = 0.83

**Table 4 jcm-13-06509-t004:** Antifungal therapy for genital candidiasis. VVC: vulvovaginal candidiasis.

Genital Infection	Treatment Recommendations
Uncomplicated acute VVC	Topical antifungal agents, with no one agent being superior to another OR a single 150 mg oral dose of fluconazole **Topical agents:** - Clotrimazole 1% (1 application per day for 7 days) or 2% (1 application per day for 5 days), OR - Clotrimazole tablets 100 mg/d for 7 days, 200 mg/d for 3 days or 500 mg single dose, OR - Miconazole 2% daily for 7 days or 1200 mg vaginal suppository; OR - Terconazole 0.8% vaginal cream daily for 3 days, OR - Nystatin cream (100,000 units/g for 7 days) is an alternative for patients allergic to imidazole
Uncomplicated acute *Candida* balanitis	Topical antifungal agents, with no one agent superior to another OR a single 150 mg oral dose of fluconazole **Topical agents:** - Clotrimazole 1% once daily for 7 days or 2% once daily for 5 days - Miconazole 2% twice daily for 7 days. - Nystatin cream 100,000 units/g for 7 days if allergy to imidazole
Severe acute VVC or balanitis	Oral fluconazole at 150 mg, given every 72 h for a total of 2 or 3 doses
*Candida glabrata* or *non albicans Candida* VVC that is unresponsive to oral azoles	- Topical intravaginal boric acid, administered in a gelatin capsule, 600 mg daily, for 14 days OR - Nystatin intravaginal suppositories, 100,000 units daily for 14 days OR - Topical 17% flucytosine cream alone or in combination with 3% Amphotericin cream administered daily for 14 days - Amphotericin B suppository 50 mg daily for 14 days
Recurrent VVC or balanitis	- Oral fluconazole at 150 mg with repeated doses on days 1, 3, and 7 - Topical agent 10–14 days of therapy, followed by oral fluconazole at 150 mg weekly for 6 months OR - Monthly oral fluconazole at 150 mg - Weekly oral fluconazole at 150 mg for 6 months, with re-evaluation at 4–6 weeks. Repeat for another 6 months if no side effects or interactions occur. Discontinue after one year. - Itraconazole 100 mg once daily orally for 6–12 months - Oteseconazole—this oral azole is FDA-approved for non-pregnant adult women. It is not approved in Europe for use in sporadic or acute VVC infections - Ibrexafungerp 300 mg orally twice a day for one day. Limit use to non-pregnant patients who prefer oral dosing but cannot use oral fluconazole (e.g., due to intolerance, allergy, or *Candida* infection resistant to fluconazole). Dose reduction required if taken with strong CYP3A inhibitors

**Table 5 jcm-13-06509-t005:** Antibacterial treatment for UTIs. IM: intramuscular; IV: intravenous; MDR: multidrug resistance; TMP-SMX: trimethoprim-sulfamethoxazole; UTI: urinary tract infection.

Treatment			
Agent	Dosage	Duration	Comment
**Acute cystitis**			
Trimethoprim–sulfamethoxazole (TMP-SMX)	One double-strength tablet (160 mg/800 mg) orally twice daily	3–7 days	Avoid if regional prevalence of resistance known to be >20%
Fosfomycin- trometahine	3 g of powder mixed in water and administered orally	Single dose	Avoid if concern for early pyelonephritis
Nitrofurantoin monohydrate/macrocrystals	100 mg orally twice daily	5–7 days	Avoid if concern for early pyelonephritis OR CrCl < 30 mL/min
Pivmecillinam	400 mg pivmecillinam orally three times daily	5–7 days	Check for beta-lactam allergy
Amoxicillin–clavulanate	500 mg orally twice daily	5–7 days	Check for beta-lactam allergy
Cefadroxil	500 mg orally twice daily	5–7 days	Check for beta-lactam allergy
Cephalexin	500 mg orally twice daily	5–7 days	Check for beta-lactam allergy
Ciprofloxacin	250 mg orally twice daily or 500 mg extended release orally once daily	3–5 days	
Levofloxacin	250 mg orally once daily	3–5 days	
**Recurrent cystitis**			
Fosfomycin- trometahine	3 g every 7 to 10 days	6 months with re-evaluation at 4–6 weeks. Repeat for another 6 months if no side effects or interactions occur	
Trimethoprim–sulfamethoxazole	40 mg/200 mg once daily OR 40 mg/200 mg three times weekly		
Trimethoprim	100 mg once daily		
Cephalexin	125 mg once daily OR 250 mg once daily		Check for beta-lactam allergy
**Pyelonefritis**			
	- Ceftriaxone 1 g IV or IM once daily OR - Ertapenem 1 g IV or IM once daily OR Use urine culture to tailor therapy Some therapies could be the following: - Amoxicillin–clavulanate 875 mg orally twice daily for 7 to 10 days or - TMP-SMX one double-strength tablet orally twice daily for 7 to 10 days or - Cefpodoxime 200 mg orally twice daily for 7 to 10 days or - Cefadroxil 1 g orally twice daily for 7 to 10 days - If low risk of fluoroquinolone resistance/toxicity: Ciprofloxacin 500 mg orally twice daily for 5 to 7 days or Levofloxacin 750 mg orally once daily for 5 to 7 days		Use ertapenem in the case of risk factors for MDR Gram-negative UTIs Any one of the following in the prior three months: - Urinary isolate isolation of previous MDR, Gram-negative (*Pseudomonas* deserves a different treatment) - Inpatient stay at a healthcare facility - Use of a fluoroquinolone, TMP-SMX, or broad-spectrum beta-lactam Avoid antibiotics with regional prevalence of resistance known to be >10% Tailor antimicrobial therapy with results of urine culture susceptibility testing results

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
