# Peer review of "Clinical Recommendations for Managing Genitourinary Adverse Effects in Patients Treated with SGLT-2 Inhibitors: A Multidisciplinary Expert Consensus"

_jcm, 2024, doi:10.3390/jcm13216509_

Round 1

Reviewer 1 Report

Comments and Suggestions for Authors

The review article on “clinical recommendations to manage genitourinary adverse effects in patients treated with sglt-2 inhibitors: a multidisciplinary expert 4 consensus” comprehensively covers the detailed information regarding the side effects associated with SGLT-2 inhibitors. The authors provide the valuable insights and guidance on strategies to prevent and manage adverse effects, particularly those affecting the genitourinary system. The review article can serve as a resource for understanding both the risks and appropriate interventions related to SGLT-2i.

I have some of the suggestions:

1.    The use of SGLT2i is associated with increased glucosuria condition leading to higher chance of urinary tract infection. As mentioned in line 238, E. coli is indeed a common uropathogens in uncomplicated UTIs, but diabetic patients are susceptible to a broader range of bacterial pathogens, and there are studies determining how presence of glucose in urine is associated with increase in their virulence characteristics. I think adding the list of the potential uropathogens causing UTIs during diabetes and how the microenvironment is favorable for pathogenicity of that uropathogens will provide thorough understanding of the topic. It is also mentioned that glucose as a nutrient source, particularly for E. coli, which seems to be not true as there are research showing that E. coli utilizes peptides and amino acids as primary source of nutrient for growth in urinary tract, while glucose might be an important source of nutrient for Proteus spp. (Proteus is mentioned in line 649 without providing any further details). Adding details with relevant citations will be helpful. 

2.    The contents on the (figure 2 and 3) and section (6.b.1 and 2) seems redundant as same information is conveyed.

3.    Some of the citations are missing (for example Line 45, 53, 205, 479, 646 and more).

Reviewer 2 Report

Comments and Suggestions for Authors

1. This is a well written and comprehensive review regarding genitourinary adverse effects associated with SGLT-2i usage. While it is unfortunate that there are not as many studies investigating the prevention and treatment of GU AEs related to SGLT-2i usage, the authors did a good job summarizing the current guidelines and recommendations of GMIs/UTI treatment.

2. The tables and figures are clear and concise.

Reviewer 3 Report

Comments and Suggestions for Authors

Dear Author,

I read with interest your article, which seems pretty complete and exhaustive

It contains many informations regarding SGLT-2 inhibitors drugs, especially regarding their side effects

It is very accurate and underlines what it's perceived by many urologists, that these drugs have a serious impact in the genitourinary tract, which need to be discussed prior with the patients

However, my only criticism is that lacks a strength and limitations sections
